# Magnitude of venous or capillary blood-derived SARS-CoV-2-specific T cell response determines COVID-19 immunity

Martin J. Scurr [1,2,3] ✉, George Lippiatt[3], Lorenzo Capitani[1,2], Kirsten Bentley [1,2], Sarah N. Lauder[1,2], Kathryn Smart[1,2], Michelle S. Somerville[1,2], Tara Rees[1,3,4], Richard J. Stanton [1,2], Awen Gallimore[1,2], James P. Hindley[3] & Andrew Godkin [1,2,3,4] ✉

T cells specific for SARS-CoV-2 are thought to protect against infection and development of COVID-19, but direct evidence for this is lacking. Here, we associated whole-blood-based measurement of SARS-CoV-2-specific interferon-γ-positive T cell responses with positive COVID-19 diagnostic (PCR and/or lateral flow) test results up to 6 months post-blood sampling. Amongst 148 participants donating venous blood samples, SARS-CoV-2-specific T cell response magnitude is significantly greater in those who remain protected versus those who become infected ($P < 0.0001$); relatively low magnitude T cell response results in a 43.2% risk of infection, whereas high magnitude reduces this risk to 5.4%. These findings are recapitulated in a further 299 participants testing a scalable capillary blood-based assay that could facilitate the acquisition of population-scale T cell immunity data (14.9% and 4.4%, respectively). Hence, measurement of SARS-CoV-2-specific T cells can prognosticate infection risk and should be assessed when monitoring individual and population immunity status.

Measuring and understanding immunological responses to SARS-CoV-2 infection is important in helping to develop effective future strategies that minimise the public health and economic impact of future outbreaks of COVID-19. Identifying correlates of immunity would provide significant insight into community susceptibility to viral infection, potentially forewarning peaks in hospitalisations, but also allow individuals to personally manage their own risk of infection to themselves and others. Immune monitoring has already proven critical to assessing the efficacy of COVID-19 vaccines in healthy and at-risk patient populations[1–3], in particular from SARS-CoV-2 mutant variants[4] and the identification of waning immunity would signify the requirement for booster vaccinations and prevent future outbreaks.

An individual's level of immunity against SARS-CoV-2 infection is dependent on multiple factors: viral load at point of exposure, viral variant, age, prior vaccination/infection status, co-morbidities, medications and most critically, the magnitude of anti-SARS-CoV-2 adaptive immune responses present at the point of viral exposure[5]. The assessment of immune responses generated to SARS-CoV-2 infection and/or vaccination has largely focussed on serological assays that measure the presence of antibodies specific for structural proteins, namely the spike glycoprotein. However, the presence or absence of antibodies alone are not an accurate determinant of protective immune responses, given both the demonstrable waning of responses over time[6] and the poor neutralising activity to SARS-CoV-2 variants in convalescent or double vaccinated individuals, which may have contributed to the large numbers of breakthrough infections[7]. Indeed, protection against symptomatic COVID-19 caused by the Omicron (B.1.1.529) variant wanes to ~10% after only 4–6 months post-mRNA

[1]Division of Infection & Immunity, School of Medicine, Cardiff University, Cardiff, UK. [2]Systems Immunity Research Institute, Cardiff University, Cardiff, UK. [3]ImmunoServ Ltd., Cardiff, UK. [4]Department of Gastroenterology & Hepatology, Cardiff & Vale University Health Board, Cardiff, UK. ✉e-mail: martinscurr@immunoserv.com; godkinaj@cardiff.ac.uk

vaccination, although protection against severe disease remained >68% for at least 7 months[8]. Measurement of adaptive memory T cell responses that enact long-term protection to viral infection may provide a better indication of susceptibility to SARS-CoV-2 infection and therefore the risk of testing COVID-19 positive[9], given that specific T cells may prevent infection without seroconversion[10,11]. However, measurement of T cell responses has received less interest due to methodological difficulties and logistical challenges of obtaining and transporting venous blood samples, especially when performing large observational studies to assess vaccine efficacy and monitor immunity. Despite this, vaccinated individuals exhibit robust T cell activity to SARS-CoV-2 variants potentially compensating loss of antibody reactivity to limit severe COVID-19[12,13].

Here, we sought to understand whether a single measurement of SARS-CoV-2 T cell response could prognosticate the absolute risk of becoming infected by SARS-CoV-2 in the 6-months following blood draw, regardless of prior factors that influence immunity. To make a T cell test high-throughput and applicable to larger population studies, we also sought to miniaturise the test to enable it to be performed using a finger prick capillary blood sample.

## Results

### Venous-blood based measurement of SARS-CoV-2 immunity

We utilised a whole venous blood-based SARS-CoV-2 T cell and IgG antibody combined assay to measure the cellular and humoral immune responses of healthy donors (see Table 1 for participant characteristics), donating blood to the COVID-Immune study between September 2021 and March 2022[14]. Amongst vaccinated donors, SARS-CoV-2-specific T cell responses, identified by measuring plasma-derived interferon-gamma (IFN-γ) following whole blood stimulation with SARS-CoV-2 peptides (as previously described refs. 14–18), and nucleocapsid (N)-binding IgG responses were elevated in those reporting prior infection, although both responses were highest in previously infected unvaccinated donors (Fig. 1a, b). IgG responses targeting the spike glycoprotein (RBD, S1, S2) were all highest in previously infected vaccinated donors (Fig. 1c–e).

Following blood sampling, participants were asked to self-report COVID-19 positive PCR and/or lateral flow test results; participants were assumed to have contracted the Delta (B.1.617.2) coronavirus variant if testing positive between 1 September 2021 and 29 December 2021, and Omicron (B.1.1.529) after 29 December 2021 when this variant of concern became dominant according to Public Health Wales. Amongst 148 evaluable donors, we observed an infection rate of 26.3% (39/148) within 6 months of blood draw, 38 of which were breakthrough infections following a second or third dose of COVID-19 vaccine (either Pfizer/BioNTech (BNT162b2) mRNA vaccine or AstraZeneca (ChAdOx1 nCoV-19) vaccine); one unvaccinated donor was also infected. The magnitude of the SARS-CoV-2-specific IFN-γ-positive T cell response was significantly lower in those reporting a positive COVID-19 diagnostic test than uninfected donors ($P < 0.0001$; Fig. 2a), predominantly due to sub-optimal induction of T cell responses by vaccination amongst certain participants ($P = 0.050$; Supplementary Fig. 1). There was no correlation between the magnitude of IFN-γ$^+$ T cell response and time prior to COVID-19-positive test result (Supplementary Fig. 2). In contrast, neither RBD-, S1-, S2-binding IgG responses (Fig. 2b–d) nor RBD-, S1-neutralising antibody responses specific for wild-type SARS-CoV-2 or the delta variant (B.1.617) (Supplementary Fig. 3) could distinguish individuals at risk of infection. However, low anti-SARS-CoV-2 N-binding IgG response did associate with risk of COVID-19 ($P = 0.0084$; Fig. 2e); indeed the odds of individuals testing COVID-19 positive were 85% smaller if they had prior confirmed history of SARS-CoV-2 infection ($P = 0.00035$; OR 0.15, 95% CI: 0.047–0.39; Supplementary Fig. 4).

Previously defined cut-offs for diagnostic positivity[14] were deemed too arbitrary for assessing risk of re-infection, hence quartile ranges were set to establish absolute risk parameters. A statistical model including only variables shown to have a significant impact on outcome revealed that the magnitude of the SARS-CoV-2-specific IFN-γ$^+$ T cell response was the most significant immunological biomarker in establishing the odds of an individual testing COVID-19 positive (Fig. 2f and Supplementary Fig. 4). Participants with SARS-CoV-2-specific IFN-γ$^+$ T cell responses in the third (194–489 pg/ml IFN-γ) and fourth (>489 pg/ml IFN-γ) quartiles had 65% ($P = 0.055$; OR 0.35, 95% CI: 0.11–1.00) and 90% ($P = 0.0050$; OR 0.098, 95% CI: 0.014–0.42) smaller odds respectively than those in the first quartile (Supplementary Fig. 4). Overall, those participants with a venous blood-derived SARS-CoV-2-specific T cell response ≤79 pg/ml IFN-γ had a 43.2% risk of breakthrough infection within 6 months, whereas those with a response >489 pg/ml IFN-γ had 5.4% risk of infection (Table 2).

## Table 1 | Participant characteristics

| Characteristic | Venous sampling (n = 148) | Capillary sampling (n = 299) |
|---|---|---|
| **Gender** | | |
| Female | 102 (68.9%) | 191 (63.9%) |
| Male | 46 (31.1%) | 108 (36.1%) |
| Age, median (IQR) | 40 (30–50) | 48 (39–60) |
| **Ethnicity** | | |
| White (British/Irish) | 107 (72.3%) | 166 (89.3%) |
| White (Other) | 4 (2.7%) | 18 (9.1%) |
| Mixed | 1 (0.7%) | 2 (1.0%) |
| Black (African) | 0 (0%) | 2 (1.0%) |
| Black (Caribbean) | 0 (0%) | 3 (1.5%) |
| Asian | 3 (2.0%) | 3 (1.5%) |
| Other (not specified) | 36 (24.3%) | 3 (1.5%) |
| **Vaccine status** | | |
| Vaccinated[a] | 143 (96.5%) | 228 (76.3%) |
| Unvaccinated | 5 (3.5%) | 71 (23.7%) |
| Weeks from COVID-19 vaccination to blood draw, median (IQR) | 7.71 (4.3–16.5) | 13.4 (9.9–16.5) |
| **Infection status** | | |
| Prior infection[b] | 64 (43.2%) | 80 (26.8%) |
| No evidence of infection | 84 (56.8%) | 219 (73.2%) |
| Weeks from positive COVID-19 test to blood draw, median (IQR) | 39.9 (9.2–82.2) | 17.4 (11.2–45.2) |
| **Co-morbidities** | | |
| None | 148 (100%) | 223 (74.6%) |
| Hypothyroid | 0 (0%) | 16 (5.4%) |
| Arthritis | 0 (0%) | 14 (4.7%) |
| Diabetes | 0 (0%) | 11 (3.7%) |
| Cancer | 0 (0%) | 9 (3.0%) |
| Colitis | 0 (0%) | 4 (1.3%) |
| Lyme disease | 0 (0%) | 3 (1.0%) |
| Lupus SLE | 0 (0%) | 2 (0.7%) |
| Osteroporosis | 0 (0%) | 2 (0.7%) |
| Other[c] | 0 (0%) | 17 (5.7%) |

[a]Denotes participants that have received at least one dose of any COVID-19 vaccine.
[b]Self-confirmed prior positive PCR and/or lateral flow test for SARS-CoV-2.
[c]Includes dermatomyositis, Hydradenitis supperativa, IgA deficiency, IgA nephropathy, hepatic encephalopathy, Hashimoto's disease, vasculitis, Graves disease, cardiopulmonary diseases, and rare genetic disorders.

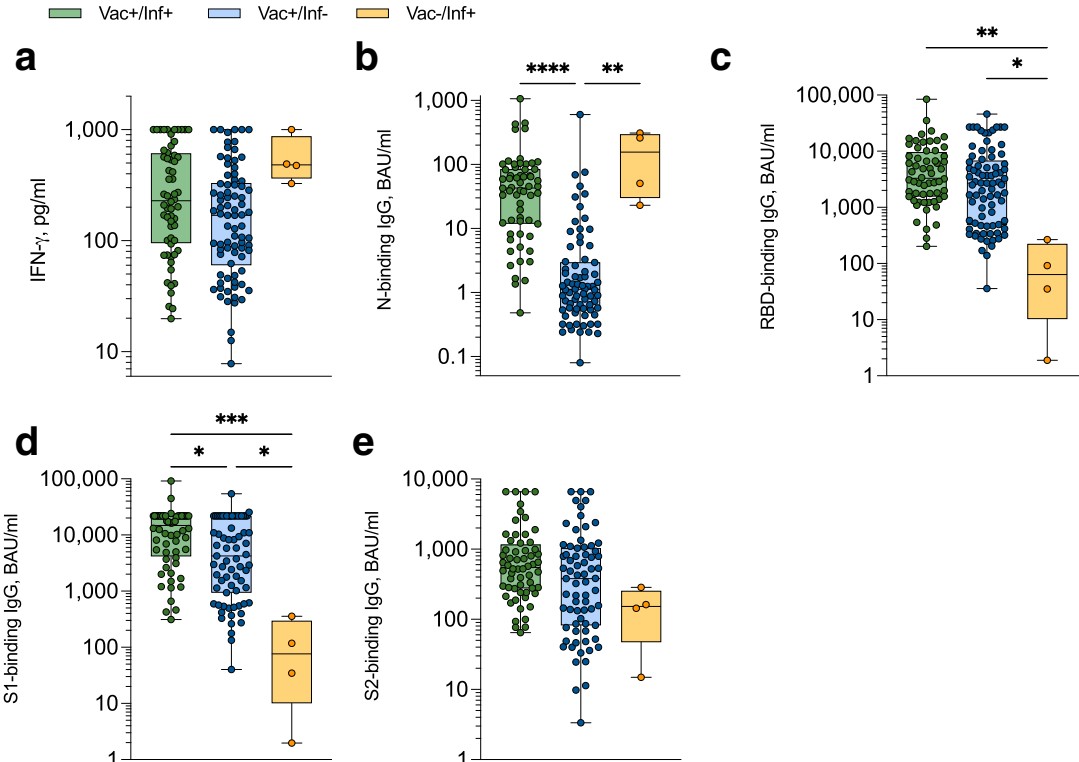

**Fig. 1 | Magnitude of anti-SARS-CoV-2 T cell and IgG responses measured in venous blood samples. a** SARS-CoV-2-specific IFN-γ+ T cell responses were measured using the venous whole blood assay and sub-divided based on participant vaccination and prior SARS-CoV-2 (PCR and/or lateral flow test confirmed) infection status. 'Vac + /Inf + ' n = 60 (green), 'Vac + /Inf-' n = 82 (blue), 'Vac-/Inf + ' n = 4 (yellow), 'Vac-/Inf-' n = 1 (not plotted). SARS-CoV-2-specific IgG-binding responses targeting nucleocapsid ('N') (**b**; ****P < 0.0001, **P = 0.0016), spike receptor binding domain ('RBD') (**c**; **P = 0.0022, *P < 0.015), spike subunit 1 ('S1') (**d**; ***P = 0.0005, *(Vac + /Inf+ vs. Vac + /Inf-) P = 0.022, *(Vac-/Inf+ vs. Vac + /Inf-) P = 0.012) and spike

subunit 2 ('S2') (**e**) were measured using the venous whole blood assay and sub-divided based on participant vaccination and prior SARS-CoV-2 (PCR and/or lateral flow test confirmed) infection status. 'Vac + /Inf + ' n = 60 (green), 'Vac + /Inf-' n = 71-82 (blue), 'Vac-/Inf + ' n = 4 (yellow). Comparisons used Kruskal-Wallis tests with correction for multiple comparisons using Dunn's tests. Data are presented as box plots (centre line at the median, upper bound at 75th percentile, lower bound at 25th percentile) with whiskers at minimum and maximum values. Each dot represents one donor. Source data are provided as a Source Data file.

## Capillary-blood based measurement of SARS-CoV-2 immunity

Venous whole blood assays are limited in scale due to the requirement for a phlebotomist to obtain samples. In order to increase accessibility of the SARS-CoV-2 T cell and IgG test, an alternative capillary blood-based sampling technique was developed, allowing participants to obtain the blood sample at home from a finger prick. To our knowledge, there are no prior reports measuring antigen-specific T cell functionality in capillary blood samples. A strong correlation between lymphocyte counts obtained using matched capillary and venous blood samples has been shown previously[19]. In addition, whole blood-based assays measuring SARS-CoV-2-specific T cell responses have been reported using as little as 320 µl of venous blood[20], thus mitigating concerns regarding precursor T cell frequency within capillary blood samples.

We utilised this high-throughput, standardised whole capillary blood-based SARS-CoV-2 T cell and IgG antibody combined assay to measure the cellular and humoral immune responses of participants with a heterogeneous range of co-morbidities and prior vaccination / infection statuses (Table 1), recruited from across the UK between 24th January and 14th March 2022[14]. The majority (90.9%) of finger-prick blood samples were obtained correctly and shipped to the laboratory within 24 h of collection. In some instances, samples arrived up to 48 h post-blood draw, though none of these samples failed quality control checks nor did this affect overall T cell or antibody measurements (Supplementary Fig. 5). Despite some individuals exhibiting variation in the magnitude of SARS-CoV-2-specific IFN-γ+ T cell responses measured in matched capillary and venous

blood samples, overall there was no significant difference (P = 0.88; Supplementary Fig. 6).

Amongst vaccinated individuals also reporting prior infection, SARS-CoV-2-specific IFN-γ+ T cell responses were significantly elevated (P = 0.0001), though not significantly higher than previously infected unvaccinated donors (P = 0.19; Fig. 3a). IgG responses targeting the spike glycoprotein (RBD, S1, S2) were all significantly higher in vaccinated versus unvaccinated donors regardless of prior infection status (Fig. 3b–d). Intriguingly, median N-binding IgG responses were highest in previously infected unvaccinated versus vaccinated participants, albeit this did not reach significance (Fig. 3e). Amongst self-reported unvaccinated and uninfected donors, 15/37 (40.5%) participants had positive N-binding IgG responses above a previously defined cut-off[14] of 2.0 BAU/ml; 12 of these 15 participants had positive IFN-γ+ T cell responses above a previously defined cut-off of 22.7 pg/ml IFN-γ[14]. As such, these participants are highly likely to have been previously infected with SARS-CoV-2 and either did not test for COVID-19 due to personal choice, lack of availability of PCR and/or lateral flow devices, or were asymptomatic. Although there was a significant correlation between IFN-γ+ T cell response and N-binding IgG level in unvaccinated donors (P = 0.0044; Supplementary Fig. 7), N-binding IgG responses waned at a greater rate in vaccinated versus unvaccinated donors, whereas IFN-γ+ T cell responses were maintained regardless of vaccination status, albeit number of donors beyond 50 weeks post-infection were low (Supplementary Fig. 8). Type of vaccination made little overall difference to observed SARS-CoV-2-specific T cell and RBD-binding IgG responses, although participants that received two doses

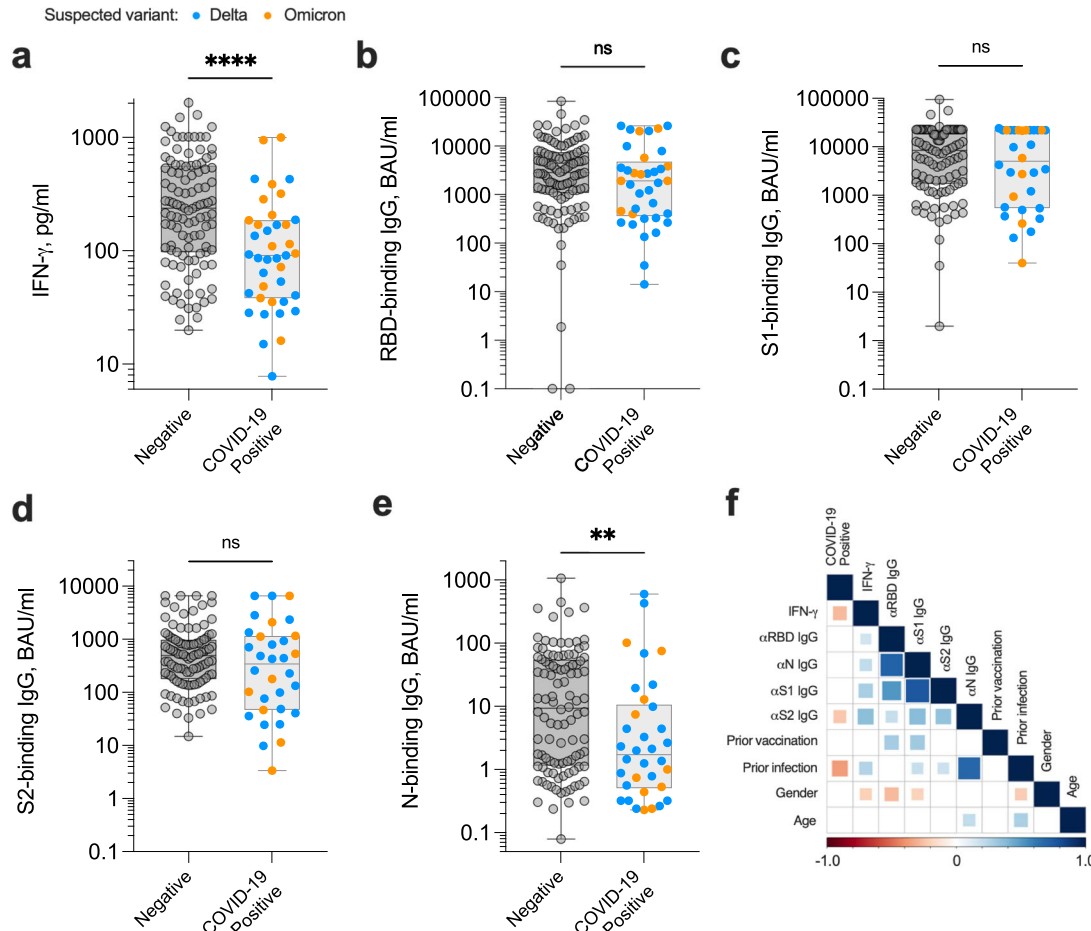

**Fig. 2 | Magnitude of anti-SARS-CoV-2 adaptive immune responses measured in venous blood samples up to six months preceding a positive COVID-19 test.** Venous blood samples obtained from healthy donors (*n* = 148) were assessed for the magnitude of SARS-CoV-2-specific IFN-γ⁺ T cell responses (**a**; ****P < 0.0001) and SARS-CoV-2-specific IgG-binding responses targeting spike receptor binding domain ('RBD') (**b**), spike subunit 1 ('S1') (**c**), spike subunit 2 ('S2') (**d**) and nucleocapsid ('N') (**e**; **P = 0.0084). Participants self-reporting a COVID-19 positive test (PCR and/or lateral flow test) are highlighted; all cases of infection occurred within 6 months of blood draw. Comparisons used two-sided Mann–Whitney tests. Data are presented as box plots (centre line at the median, upper bound at 75th percentile, lower bound at 25th percentile) with whiskers at minimum and maximum values. Each dot represents one donor. ns not significant. **f** Heat map demonstrating Spearman's rank correlations between specified dataset variables. Comparisons that were not statistically significant were excluded from the matrix and are represented by empty boxes. Source data are provided as a Source Data file.

of BNT162b2 followed by an mRNA1273 booster exhibited significantly higher magnitude IFN-γ⁺ T cell responses to SARS-CoV-2 than those that received two doses of ChAdOx1 followed by BNT162b2 (Supplementary Fig. 9). In addition, reported co-morbidities made little overall

difference to observed T cell responses when compared to otherwise heathy donors (Supplementary Fig. 10).

As previously, participants were asked to self-report COVID-19 positive PCR and/or lateral flow test results; participants were assumed to have contracted the Omicron (B.1.1.529) coronavirus variant when testing positive, given that it was the dominant variant across the UK during the study time period according to the UK Health Security Agency. Amongst 299 evaluable donors, we observed an infection rate of 8.0% (24/299) within three months of capillary blood donation, seven of whom were unvaccinated. The presence of co-morbidities as a proportion of all participants was lower amongst participants testing COVID-19 positive (10.7%) than those remaining negative (24.4%, Table 1), a potential result of increased caution and shielding amongst participants with certain diseases such as diabetes and cancer. As observed amongst the venous blood cohort, the magnitude of the SARS-CoV-2-specific interferon-gamma (IFN-γ)-positive T cell response measured in capillary blood samples was significantly lower in those reporting a positive COVID-19 diagnostic test than uninfected donors (*P* = 0.034; Fig. 4a), due to relatively poor induction of T cell responses by vaccination and/or prior infection (Supplementary Fig. 11). Again, neither RBD-, S1-, S2-binding IgG responses (Fig. 4b–d) nor RBD-, S1-neutralising antibody responses specific for wild-type SARS-CoV-2 or

**Table 2 | Absolute risk of SARS-CoV-2 infection within 6 months of blood sampling**

| Blood source | Quartile | SARS-CoV-2-specific T cell response IFN-γ, pg/ml | COVID-19+ <6 months (vein) <3 months (capillary) | Infection rate (%) |
|---|---|---|---|---|
| Vein | Q1 (0–25%) | <79 | 16/37 | 43.2 |
| | Q2 (25–50%) | 79–194 | 15/37 | 40.5 |
| | Q3 (50–75%) | 194–489 | 6/37 | 16.2 |
| | Q4 (75–100%) | >489 | 2/37 | 5.4 |
| Capillary | Q1 (0–25%) | <23.7 | 10/67 | 14.9 |
| | Q2 (25–50%) | 23.7-58.7 | 6/68 | 8.8 |
| | Q3 (50–75%) | 58.7-141.6 | 5/67 | 7.5 |
| | Q4 (75–100%) | >141.6 | 3/68 | 4.4 |

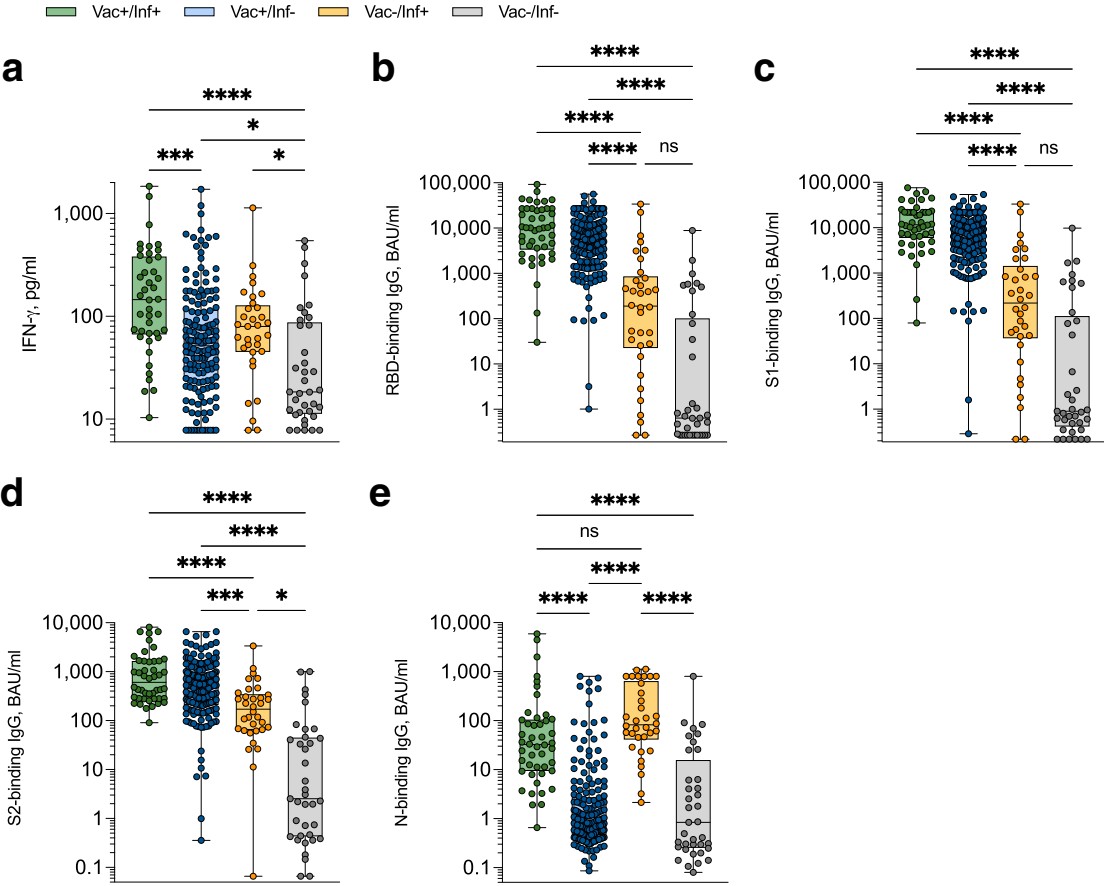

**Fig. 3 | Magnitude of anti-SARS-CoV-2 T cell and IgG responses measured in capillary blood samples. a** SARS-CoV-2-specific IFN-γ⁺ T cell responses were measured using the capillary whole blood assay and sub-divided based on participant vaccination and prior SARS-CoV-2 (PCR and/or lateral flow test confirmed) infection status. 'Vac + /Inf + ' n = 42 (green), 'Vac + /Inf-' n = 158 (blue), 'Vac-/Inf + ' n = 33 (yellow), 'Vac-/Inf-' n = 37 (grey). ****P < 0.0001, ***P = 0.0001, *(Vac + /Inf- vs. Vac-/Inf-) P = 0.045, *(Vac-/Inf+ vs. Vac-/Inf-) P = 0.014. SARS-CoV-2-specific IgG-binding responses targeting spike receptor binding domain ('RBD') (**b**; ****P < 0.0001, ns: not significant), spike subunit 1 ('S1') (**c**; ****P < 0.0001, ns: not significant), spike subunit 2 ('S2') (**d**; ****P < 0.0001, ***P = 0.0005, *P = 0.016) and

nucleocapsid ('N') (**e**; ****P < 0.0001, ns not significant) were measured using the venous whole blood assay and sub-divided based on participant vaccination and prior SARS-CoV-2 (PCR and/or lateral flow test confirmed) infection status. 'Vac + /Inf + ' n = 46 (green), 'Vac + /Inf-' n = 182 (blue), 'Vac-/Inf + ' n = 34 (yellow), 'Vac-/Inf-' n = 37 (grey). Comparisons used Kruskal-Wallis tests with correction for multiple comparisons using Dunn's tests. Data are presented as box plots (centre line at the median, upper bound at 75th percentile, lower bound at 25th percentile) with whiskers at minimum and maximum values. Each dot represents one donor. Source data are provided as a Source Data file.

the delta variant (B.1.617) (Supplementary Fig. 12) could distinguish individuals at risk of infection with any degree of significance. Unlike the venous cohort, N-binding IgG responses also did not distinguish risk of COVID-19 (Fig. 4e), highly suggestive of increased immune evasion by the Omicron (B.1.1.529) variant in previously infected individuals, as recently described[21]. Instead, the magnitude of the SARS-CoV-2-specific IFN-γ T cell response was again the most significant variable in establishing the odds of an individual testing COVID-19 positive (Fig. 4f). Overall, those participants with a capillary blood-derived SARS-CoV-2-specific T cell response ≤23.7 pg/ml IFN-γ had a 14.9% risk of infection within three months, whereas those with a response >141.6 pg/ml IFN-γ had 4.4% risk of infection (Table 2).

## Discussion

As we enter the next phase of the COVID-19 pandemic, the emphasis is switching from prevention to personalised risk management and identification of vulnerable members of society. Establishing correlates of immunity to COVID-19 are critical to effectively identify and manage such at-risk individuals. There is now increasing evidence demonstrating a protective role of T cell immunity in both preventing SARS-CoV-2 infection and limiting COVID-19 severity[10]. The data presented here demonstrate that the combined magnitude of SARS-CoV-

2-specific IFN-γ⁺ T cell responses directed towards the spike, membrane and nucleocapsid structural proteins, is a better correlate of protection against developing COVID-19 than antibody-binding or -neutralising responses, and must be considered when assessing individual and/or population immunity. RNA viruses, such as SARS-CoV-2 or influenza A virus (IAV), escape serological neutralisation by rapid evolution of exposed B cell epitopes in surface antigens that are recognised by antibodies. The protective immune response offered by T cells is likely to reflect the targeting of epitopes derived from more conserved regions of viral proteins that do not rapidly escape the immune response. The T cell mediated protection to emerging SARS-CoV-2 variants is analogous to the heterosubtypic protection seen in IAV subtypes mediated by T cells targeting conserved internal proteins[22,23].

Despite the profound potential for measuring cellular immune responses to COVID-19, relatively little attention has been given to the development of accurate, high-throughput, standardisable T cell tests. Traditional complexities and costs associated with measuring T cell responses have hindered accurate determinations of T cell immunity in large population cohort immunity screens. Whilst some commercial whole blood peptide-stimulation assays have recently come to market, all still currently require a phlebotomist to obtain blood, limiting

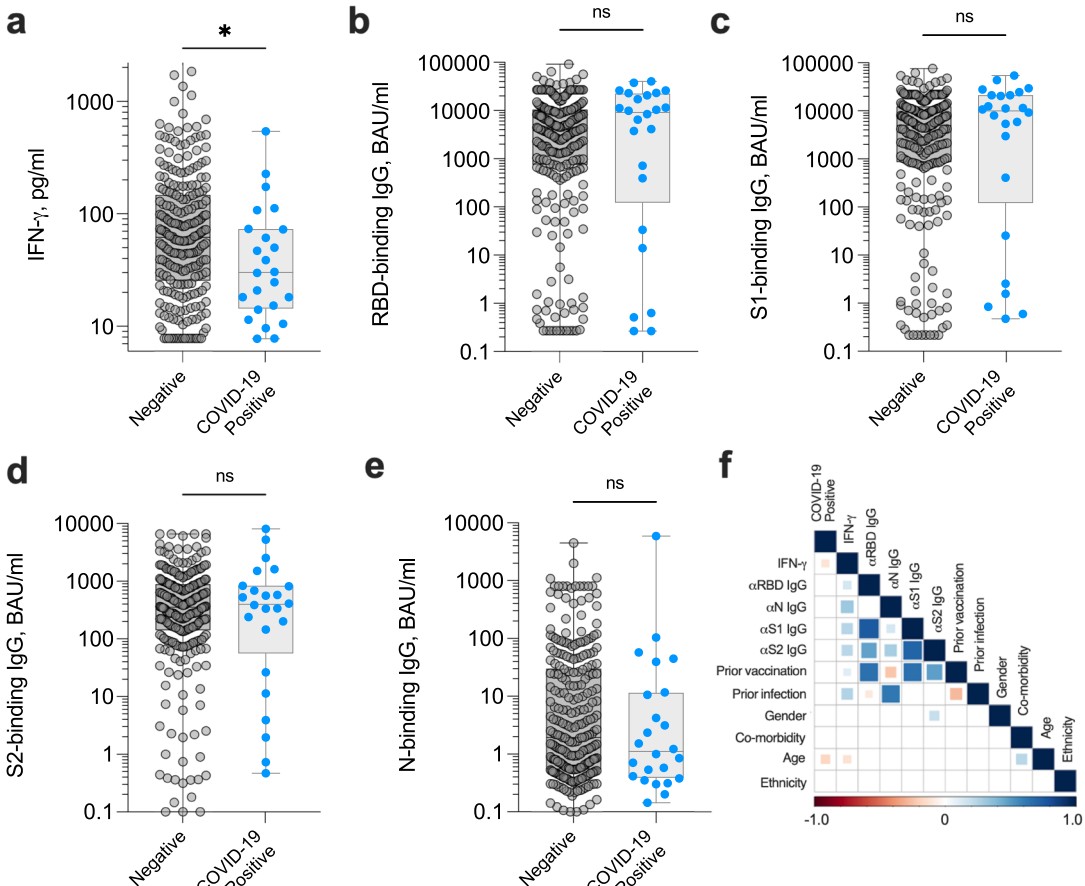

**Fig. 4 | Magnitude of anti-SARS-CoV-2 adaptive immune responses measured in capillary blood samples up to 3 months preceding a positive COVID-19 test.** Capillary blood samples obtained from study participants ($n = 299$) were assessed for the magnitude of SARS-CoV-2-specific IFN-γ⁺ T cell responses (**a**; *$P = 0.034$) and SARS-CoV-2-specific IgG-binding responses targeting spike receptor binding domain ('RBD') (**b**), spike subunit 1 ('S1') (**c**), spike subunit 2 ('S2') (**d**) and nucleo-capsid ('N') (**e**). Participants self-reporting a COVID-19 positive test (PCR and/or lateral flow test) are highlighted; all cases of infection occurred within 3 months of blood draw. Comparisons used two-sided Mann–Whitney tests. Data are presented as box plots (centre line at the median, upper bound at 75th percentile, lower bound at 25th percentile) with whiskers at minimum and maximum values. Each dot represents one donor. ns not significant. **f** Heat map demonstrating Spearman's rank correlations between specified dataset variables. Comparisons that were not statistically significant were excluded from the matrix and are represented by empty boxes. Source data are provided as a Source Data file.

accessibility and scale. Capillary blood based systems are already in widespread use for determining SARS-CoV-2 antibody prevalence in the population[24]. We adapted the capillary blood test to perform a whole blood peptide-stimulation assay to assess both T cell reactivity to SARS-CoV-2 structural proteins and SARS-CoV-2 specific antibody responses. Indeed, combined measurements of SARS-CoV-2-specific antibody and T cells in the same capillary blood sample is highly attractive: (i) reduces the requirement for multiple blood tests from each participant; (ii) improves participant experience and uptake; (iii) improves logistics and reduces duplication, and; (iv) reduces environmental impact, given fewer laboratory consumables and sample deliveries are required. Whilst overall IFN-γ responsiveness was equivalent between matched venous and capillary blood-derived samples, lower overall median IFN-γ values were observed in the capillary blood participant cohort (Fig. 4a) in comparison to the venous blood cohort (Fig. 2a). Several explanations may account for this finding; namely a larger number of participants with co-morbidities requiring immunosuppressive treatments were recruited to the capillary blood sampling cohort (Table 1), and the viability and/or functionality of T cells obtained from capillary samples may be lower, especially given the prolonged storage condition of the sample prior to peptide stimulation.

The COVID-19 vaccines currently in widespread use provide optimal protection from severe illness within 6 months of dosage for the majority of recipients[8]. It is encouraging to note that in spite of poor vaccine-induced serological neutralisation of SARS-CoV-2 variants[6,7], T cell responses induced by vaccination to wild-type SARS-CoV-2 maintained strong reactivity to delta and omicron variants, as shown by others[25]. Our data presented here demonstrates the importance of assessing vaccine immunogenicity more broadly, highlighting those with insufficient T cell immunity to prevent breakthrough infection and continued viral transmission. Regardless of prior vaccination, we also observed many instances of unvaccinated individuals recruited to the capillary cohort study with sizeable SARS-CoV-2-specific T cell (and N-binding IgG) responses, most likely from prior infection. Rather than imposing vaccination on concerned individuals, an assessment of their risk of infection could be based on current immunity status, and informed choices made.

Limitations of this study include the reliance on participants self-reporting SARS-CoV-2 infection after blood sampling to identify correlates of immunity; it is possible some participants exhibited asymptomatic infections and did not perform a PCR and/or lateral flow test for COVID-19. Our dataset also lacked information regarding participant medication at time of blood draw. Furthermore, immune responses that prognosticate severe disease and increased risk of hospitalisation with COVID-19 could not be identified from our dataset, given that all of our participants reported only mild/moderate symptoms or were asymptomatic. However, the presence of CD8⁺ T cell

responses directed towards a nucleocapsid-specific epitope have recently been correlated with protection from severe COVID-19[26]. In addition, the test utilised here did not measure T cell responses to particular early expressed non-structural SARS-CoV-2 proteins that were recently shown to preferentially accumulate in seronegative healthcare workers exposed to infectious patients[27]. In accordance with this work, the magnitude of SARS-CoV-2-specific T cells identified in our test also appear capable of clearing subclinical infection, given the prevalence of community transmission at the time of recruitment and the high likelihood of exposure to infection amongst our cohorts. Finally, we did not measure interleukin-2 production by T cells given our prior work demonstrating inferior identification of SARS-CoV-2-specific T cell responses[14], although IL-2-specific response could be indicative of pre-existing, cross-reactive memory T cells which have been associated with protection from SARS-CoV-2 infection[11].

In conclusion, these data emphasise the essential requirement for long-term longitudinal studies that incorporate SARS-CoV-2-specific T cell responses for population-scale measurements of immunity. The development of a novel capillary blood test that measures T cell responses will likely facilitate such an endeavour.

## Methods

### Participants

Participants were recruited to this research project between February 2021 and March 2022. A healthy donor cohort donating venous blood samples ($n = 148$) primarily comprised university staff and students attending Cardiff University's COVID-19 Screening Service or primary school staff at a Cardiff-based school. All participants were otherwise healthy and did not report taking any current immunosuppressive medication (see Table 1 for characteristics). The participant cohort donating capillary blood samples comprised any willing donor (over 18 years old) from across the United Kingdom. A total of 342 participants were recruited to the study between 24th January and 14 March 2022, with 299 participants returning blood samples to the laboratory. Many participants remained unvaccinated and/or reported significant co-morbidities, including autoimmune diseases and cancer (see Table 1 for characteristics). This study received ethical approval from the Newcastle & North Tyneside 2 Research Ethics Committee (IRAS ID: 294246) and Cardiff University School of Medicine Research Ethics Committee (SREC reference: SMREC 21/01). All participants gave written, informed consent prior to inclusion. Participants did not receive any compensation for their involvement in this study.

### Blood sampling

Venous blood samples were obtained by venepuncture into 6 or 10 ml lithium or sodium heparin vacutainers (BD). Capillary blood samples were obtained by lancet blade incision of a finger and subsequent collection into a heparin microtainer (BD). A minimum of 400 µl blood was required; any sample underfilled below this amount was rejected. Other reasons for sample rejection included profuse clotting and/or haemolysis and viscous plasma that could not be collected for assays (Supplementary Fig. 5). Overall, 299 capillary blood samples were evaluable for antibody responses; of these, 270 samples were also evaluable for T cell responses.

### Stimulation

SARS-CoV-2-specific T cell responses were assessed using the COVID-19 Immuno-T test (ImmunoServ Ltd), and performed as previously described[14]. Briefly, a single 6-ml or 10-ml sodium heparin vacutainer (BD) tube of venous blood was collected from each participant and processed in the laboratory within 12 h of blood draw. A single 400–600 µl heparin microtainer (BD) tube of capillary blood was collected from participants donating a finger prick blood sample within 48 h of blood draw, although majority of samples were processed within 24 h. Venous and/or capillary blood samples were stimulated with a single SARS-CoV-2 (wild-type variant)-specific peptide pool, as previously described[14]. This peptide pool comprised 420 15-mer sequences with 11 amino acid overlap, covering the entire spike (S1 and S2) protein (S; NCBI Protein: QHD43416·1), nucleocapsid phosphoprotein (NP; NCBI Protein: QHD43423·2) and membrane glycoprotein (M; NCBI Protein: QHD43419·1) coding sequences (termed 'S-/NP-/M-combined peptide pool'). All peptides were purified to >70%, dissolved in sterile water and used at a final concentration of 0.5 µg/ml per peptide. Samples were incubated at 37 °C for 20–24 h. Tubes were then centrifuged at 5000×$g$ for 3 min before harvesting ~150 µl plasma from the top of each blood sample. Plasma samples were stored at −20 °C for up to one month prior to running cytokine / antibody detection assays.

### IFN-γ T cell assay

IFN-γ was measured using the IFN-γ ELISA MAX Deluxe Set (BioLegend, catalogue #430116) and performed according to the manufacturer's instructions. Immediately following the addition of stop solution (2 N $H_2SO_4$), microplates were read at 450 nm using the BioLegend Mini ELISA Plate Reader. IFN-γ was quantified by extrapolating from the standard curve using GraphPad Prism. Values below the lower limit of detection of the assay were recorded as 7.8 pg/ml; values above the upper limit of detection of the assay were recorded as 1000 pg/ml.

### Anti-SARS-CoV-2 antibody assay

Anti-SARS-CoV-2 RBD/S1/S2/N IgG antibodies were measured using the Bio-Plex Pro Human IgG SARS-CoV-2 4-plex panel (Bio-Rad, catalogue #12014634) and performed according to the manufacturer's instructions. Samples that recorded values above the limit of quantification were re-run at 1:1000 dilution. The mean fluorescent intensity of the beads was measured on a Bio-Plex 200 (Bio-Rad). Antibody concentration was calculated by performing the assay with the VIROTROL SARS-CoV-2 single-level control (Bio-Rad), then converted to WHO/NIBSC 20/136 international reference standard units (BAU/ml) using manufacturer calibration factors.

### Anti-SARS-CoV-2 antibody neutralisation assay

Anti-SARS-CoV-2 neutralising antibodies specific for RBD and S1 subunits of wild-type and delta (B.1.617) SARS-CoV-2 lineages were measured using the Bio-Plex Pro Human SARS-CoV-2 Variant Neutralisation Antibody kit (Bio-Rad, catalogue #12016897), according to the manufacturer's instructions. Mean fluorescence intensities were measured on a Bio-Plex 200 (Bio-Rad) and percentage inhibition (i.e. neutralisation) was calculated using the following formula:

$$\text{Percentage inhibition} = (1 - [\text{MFI of sample/MFI of negative control}]) \times 100$$

### SARS-CoV-2 neutralisation assay

Assays for infectious SARS-CoV-2 neutralisation were carried out as previously described[28]. Briefly, 600PFU of wild-type SARS-CoV-2 was incubated with 3-fold serial dilutions of plasma in duplicate for 1 h at 37 °C. The mixes were then added to VeroE6 cells for 48 h. Monolayers were fixed with 4% paraformaldehyde, permeabilised with 0.5% NP-40, then incubated in blocking buffer (PBS containing 0.1% Tween and 3% non-fat milk) for 1 h. Primary antibody (anti-nucleocapsid 1C7, Stratech) was added in blocking buffer for 1 h at room temperature. After washing, secondary antibody (anti-mouse IgG-HRP, Pierce) was added in blocking buffer for 1 h. Monolayers were washed, developed using Sigmafast OPD, and read on a Clariostar Omega plate reader. Wells containing no virus, virus but no antibody, and a standardised serum displaying moderate activity were included as controls in every experiment.

## Statistics

Statistical analyses were performed in GraphPad Prism (version 9.4.1). Dataset normality was tested using the Shapiro–Wilk test. Non-parametric tests were used for all comparisons. Mann–Whitney tests were used for unmatched samples. All tests were performed two-sided with a nominal significance threshold of $P \leq 0.05$.

Initial exploratory analysis of the datasets was performed in R (version 4.0.3). This encompassed the development of univariate Spearman's rank correlation matrix, in which the correlation between two variables was represented by the size and colour of the square. The statistical significance between associations was computed using Spearman's rho, where a value $\leq 0.05$ was considered significant. Comparisons that were not statistically significant were excluded from the matrix and are represented by empty boxes. $P$-values were adjusted for multiple comparisons using Holm's correction. A binary logistic regression model was employed to model the impact of variables within out dataset on COVID-19 positivity. IFN-γ T cell response and anti-RBD/S1/S2/N IgG titre metrics were converted into factors, in which each individual was classified into their corresponding quartile for each metric. Following this, an initial exploratory model was developed using the glm function from the stats package (V4.0.3). The odds ratios derived from this initial model were extracted from model coefficients using the 'odds_plot' function from the OddsPlotty package (V1.0.2). In the development of a cross-validated model, we employed the 'bestglm' function from the bestglm package (V0.37.3) to limit user bias and enable the selection of the best predictive variable subset. The method selected was 'exhaustive' and the information criterion used to evaluate model fitting was AIC. The same workflow described above was used to derive the odds ratios.

## Reporting summary

Further information on research design is available in the Nature Research Reporting Summary linked to this article.

## Data availability

## Code availability

R code used for the creation of statistical modelling is openly available without request[29].

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

## Acknowledgements

The authors are grateful to Dr Zack Saud (Cardiff University) for assistance with performing SARS-CoV-2 neutralisation assays, Sarah Morgan (Birchgrove Primary School) for facilitating blood sample acquisition and Prof Miles Davenport (University of New South Wales) for useful discussion and critical appraisal of this manuscript. The authors extend their thanks to all participants involved with this study. This work was funded by an Innovate UK/Small Business Research Initiative award ('Assays for SARS-CoV-2 cellular immune responses' (10007867) to MJS), a UKRI COVID-19 National Core Study Immunity programme grant ('SARS-CoV-2 Optimal Cellular Assays' to AnG) and a Medical Research Council grant ('A UK underpinning platform to study immunology and immunopathology of COVID-19: The UK Coronavirus Immunology Consortium' (MR/V028448/1) to RJS and AwG). AnG and AwG are additionally supported by grants from the Wellcome Trust (209213/Z/17/Z), Cancer Research Wales and Cancer Research UK (C16731/A21200). RJS is additionally supported by the Medical Research Council (MR/S00971X/1) and Accelerate Wales. The funders had no role in study design, data collection and analysis, decision to publish or preparation of the manuscript. For the purpose of Open Access, the author has applied a CC BY public copyright licence to any Author Accepted Manuscript version arising from this submission.

## Author contributions

Conceptualisation: M.J.S., J.P.H. and An.G. Methodology or acquisition of data: M.J.S., G.L., L.C., K.B., S.N.L., K.S., M.S.S. and T.R. Analysis or interpretation of data: M.J.S., G.L., L.C., K.B., S.N.L., T.R., R.J.S., Aw.G., J.P.H. and An.G. Funding acquisition: M.J.S., J.P.H. and An.G. Project administration, technical or material support: G.L., K.S., M.S.S. and T.R. Supervision: M.J.S., Aw.G., J.P.H. and An.G. Writing – original draft: M.J.S., J.H. and An.G. Writing – review and editing: M.J.S., R.J.S., Aw.G., J.P.H. and An.G.

## Competing interests

M.J.S and An.G are founders of and hold equity in ImmunoServ Ltd. J.P.H. holds equity in ImmunoServ Ltd. G.L. and T.R. are employees of ImmunoServ Ltd. All other authors declare no competing interests.
