## [Peer review file · Nature Communications]

REVIEWERS' COMMENTS

Reviewer #2 (Remarks to the Author):

In this revised version of this manuscript, Scurr et al addressed most of the comments I previously had on the manuscript to satisfaction. However, I have some comments the authors should address in their revised version:

Major comments:

- The title is really strong in wording. Indeed, the authors find that virus-specific T-cells measured in this cohorts with these assays are most associated with preventing COVID-19. However, do the authors feel that making a generalized statement that T-cells by themselves determine immunity is warranted?
- Line 244-245: This is a really important limitation. Since PCR or lateral flows will mainly have been performed upon symptoms, the authors might have missed asymptomatic infections in their follow-up. I think this is important enough to specifically state in this paragraph of the discussion.
- Line 80, line 94-95: these conclusions are dependent on 4 dots in the dataset. I think the wording is too strong, for such an underpowered observation. In line 94/95, I don't see why the authors conclude that vaccination leads to a 'poor' T-cell response. Additionally, there were 5 unvaccinated donors in the dataset, why is there only data for 4 of them in the main figure? These statements are also not backed up by the capillary dataset.

Minor comments:

- There are many wordings the authors should correct. "SARS-CoV-2 virus" virus should be SARS-CoV-2, "COVID-19 disease" should be COVID-19. "Protective immunity" should be either "immunity" or "protective immune responses".
- Line 20-22 reads difficult, almost like that the presence of T-cells correlated with a positive PCR, I would rephrase.
- The authors have not included a single reference in line 33-41, the first paragraph in their introduction. Line 38-41 definitely warrants referencing.
- In line 47-49, the authors directly state that low neutralization of variants is responsible for a large number of breakthrough infections. I think the two are definitely associated, but causality was not determined in reference 2. The authors should weaken that statement, or remove the last part of the sentence.
- Line 72: consider adding the wording 'venous sampling' somewhere in the title, to better distinguish the two methods.

- Line 145-147: statement is not backed up by data: vac+/inf+ and vac-/inf+ are comparable for N-specific antibodies, and this should be stated in the results.
- Line 153: remove wording 'strong'. There is correlation, but I don't see any backup why this correlation is 'strong'. Consider using significant instead.
- Line 154-158: again, this waning is skewed by 4-5 donors that were followed longer than ± 25 weeks. I don't think the authors should draw strong conclusions from this.
- Line 181-183: speculation should be done in the discussion.
- Line 225-227: I find this a weak line of argumentation, the authors themselves showed that the comorbidities they included had little effect in supplemental figure 10.
- Reconsider the wording "live" SARS-CoV-2. Rather use "infectious".

Reviewer #3 (Remarks to the Author):

Authors have responded adequately to my comments.

Reviewer 2

In this revised version of this manuscript, Scurr et al addressed most of the comments I previously had on the manuscript to satisfaction. However, I have some comments the authors should address in their revised version:

Major comments:

- *The title is really strong in wording. Indeed, the authors find that virus-specific T-cells measured in this cohorts with these assays are most associated with preventing COVID-19. However, do the authors feel that making a generalized statement that T-cells by themselves determine immunity is warranted?*

Whilst we agree the title is strong in wording, it accurately communicates the main findings and hence novelty of the manuscript. Furthermore, any searches for ‘T cell immunity’ and ‘SARS-CoV-2’ will quickly identify this paper from the title.

- *Line 244-245: This is a really important limitation. Since PCR or lateral flows will mainly have been performed upon symptoms, the authors might have missed asymptomatic infections in their follow-up. I think this is important enough to specifically state in this paragraph of the discussion.*

A new section has been added to the Discussion to address this point (page 8, lines 248-250).

- *Line 80, line 94-95: these conclusions are dependent on 4 dots in the dataset. I think the wording is too strong, for such an underpowered observation. In line 94/95, I don’t see why the authors conclude that vaccination leads to a ‘poor’ T-cell response. Additionally, there were 5 unvaccinated donors in the dataset, why is there only data for 4 of them in the main figure? These statements are also not backed up by the capillary dataset.*

These conclusions are actually based solely on the vaccinated uninfected participants, whereby only those that go on to test positive for COVID-19 exhibit significantly lower vaccine-induced T cell responses (P=0.050). No comparison was made between the vaccinated and unvaccinated cohorts, due to the insufficient number of samples, as pointed out by the reviewer. However, given this confusion, we have amended this section of the results to temper the conclusion and clarify that vaccination induces sub-optimal T cell responses only in a certain minority of participants (page 3, lines 94-95).

One unvaccinated participant was also not previously infected, so cannot be grouped with the four other unvaccinated participants who were previously infected. Given that no definite conclusion can be derived from the data generated by one participant, we did not plot this result on graphs that group participants based on prior infection and vaccination status (Figure 1 and Extended Data Figure 1). For clarity, we have added a section to the legends of Figure 1 (page 18, line 518) and Extended Data Figure 1 to state that this participant was not plotted.

Minor comments:

- *There are many wordings the authors should correct. “SARS-CoV-2 virus” virus should be SARS-CoV-2, “COVID-19 disease” should be COVID-19. “Protective immunity” should be either “immunity” or “protective immune responses”.*

These have been corrected throughout the manuscript (page 1, line 34; page 2, line 49; page 6, line 205).

- *Line 20-22 reads difficult, almost like that the presence of T-cells correlated with a positive PCR, I would rephrase.*

This sentence has been amended (page 1, lines 20-21).

- *The authors have not included a single reference in line 33-41, the first paragraph in their introduction. Line 38-41 definitely warrants referencing.*

New references have been included (page 2, lines 38-40).

- *In line 47-49, the authors directly state that low neutralization of variants is responsible for a large number of breakthrough infections. I think the two are definitely associated, but causality was not determined in reference 2. The authors should weaken that statement, or remove the last part of the sentence.*

This sentence has been amended to temper the statement (page 2, line 51).

- *Line 72: consider adding the wording 'venous sampling' somewhere in the title, to better distinguish the two methods.*

Result sub-title has been amended (page 2, line 72).

- *Line 145-147: statement is not backed up by data: vac+/inf+ and vac-/inf+ are comparable for N-specific antibodies, and this should be stated in the results.*

Figure 3e clearly shows that N-binding IgG responses were highest in previously infected unvaccinated versus vaccinated participants. However, for the avoidance of doubt, we have added 'median' N-binding IgG responses to the sentence (page 5, line 146).

- *Line 153: remove wording 'strong'. There is correlation, but I don't see any backup why this correlation is 'strong'. Consider using significant instead.*

Wording has been amended (page 5, line 154).

- *Line 154-158: again, this waning is skewed by 4-5 donors that were followed longer than ± 25 weeks. I don't think the authors should draw strong conclusions from this.*

This sentence has been amended to clarify the relatively low numbers of donors beyond 50 weeks post-infection (page 5, line 158).

- *Line 181-183: speculation should be done in the discussion.*

Whilst we accept this sentence ends with a speculative comment, we believe it is important to retain this statement here to reiterate the fact that the COVID-19 positive capillary cohort participants were most likely infected with the SARS-CoV-2-Omicron variant.

• *Line 225-227: I find this a weak line of argumentation, the authors themselves showed that the comorbidities they included had little effect in supplemental figure 10.*

Our study was not designed to assess the varying immune responses amongst participants with a wide range of comorbidities; as such, the fact that the capillary cohort group contained far more participants with comorbidities (versus the healthy donor venous cohort) should at least be discussed as a possible contributing factor to explain the reduced immune responses seen in the capillary cohort.

• *Reconsider the wording “live” SARS-CoV-2. Rather use “infectious”.*

Wording has been updated (page 11, line 347).